# Age-Related Changes in Left Ventricular Vortex Formation and Flow Energetics

**DOI:** 10.3390/jcm10163619

**Published:** 2021-08-16

**Authors:** Jeffrey Shi Kai Chan, Dawnie Ho Hei Lau, Yiting Fan, Alex Pui-Wai Lee

**Affiliations:** 1Division of Cardiology, Department of Medicine and Therapeutics, Prince of Wales Hospital, The Chinese University of Hong Kong, Hong Kong, China; skjchan@link.cuhk.edu.hk (J.S.K.C.); 1155067981@link.cuhk.edu.hk (D.H.H.L.); 2Laboratory of Cardiac Imaging and 3D Printing, Li Ka Shing Institute of Health Sciences, Hong Kong, China; 3Department of Cardiology, Shanghai Chest Hospital, Shanghai Jiao Tong University, 241 West Huaihai Road, Xu Hui District, Shanghai 200030, China; fyt2450@shchest.org

**Keywords:** echocardiography, vector flow mapping, intracardiac vortex, aging

## Abstract

Analysis of the cardiac vortex has been used for a deeper understanding of the pathophysiology in heart diseases. However, physiological changes of the cardiac vortex with normal aging are incompletely defined. Vector flow mapping (VFM) is a novel echocardiographic technique based on Doppler and speckle tracking for analysis of the cardiac vortex. Transthoracic echocardiography and VFM analysis were performed in 100 healthy adults (33 men; age = 18–67 years). The intracardiac flow was assessed throughout the cardiac cycle. The size (cross-sectional area) and circulation (equivalent to the integral of normal component of vorticity) of the largest vortices in systole (S-vortex), early diastole (E-vortex), and late diastole (A-vortex) were measured. Peak energy loss (EL) was calculated from information of the velocity vector of intracardiac flow in systole and diastole. With normal aging, the circulation (*p* = 0.049) of the E-vortex decreased, while that of the A-vortex increased (both *p* < 0.001). E-vortex circulation correlated directly to e’ (*p* = 0.003), A-vortex circulation correlated directly to A and a’ (both *p* < 0.001), and S-vortex circulation correlated directly to s’ (*p* = 0.032). Despite changes in vortex patterns, energy loss was not significantly different in older individuals. Normal aging is associated with altered intracardiac vortex patterns throughout the cardiac cycle, with the late-diastolic A-vortex becoming physiologically more dominant. Maintained energy efficiency accompanies changes in vortex patterns in aging hearts.

## 1. Introduction

Vortices are rotational whirling bodies of fluid. In recent decades, advances in imaging techniques including cardiac magnetic resonance, particle image velocimetry [1], and vector flow mapping (VFM) have enabled in vivo analysis of intracardiac vortices. VFM is an echocardiographic technique to visualize intracardiac blood flow by combining flow vector components acquired in color Doppler images and wall motion information obtained by speckle tracking. Such a combination of information from color flow Doppler and speckle tracking echocardiography overcomes the angle dependency inherent to traditional Doppler techniques [2]. It can reproducibly generate parameters that reflect intracardiac hemodynamics, including energy loss (EL), which is the energy dissipated via the frictional heat generated due to the viscosity of blood at sites of turbulent flow [3,4]. Data that are directly descriptive of the intracardiac vortex can also be generated, including the vortex area and vortex circulation—a summation of all flow velocity components in a vortex, thereby reflecting both the direction (with negative values indicating clockwise direction) and intensity of a vortex. Given that intracardiac vortices facilitate left ventricular (LV) filling and redirect blood flow for LV ejection [5,6,7], VFM may provide incremental information on LV function when compared to conventional methods [8].

Aging is associated with a reduction in LV suction force and an increase in dependence on atria-driven active LV filling [9]. While early reports have shown that transmitral flow velocity independently predicts the end-diastolic vortex area [10,11] and that intracardiac energetics may be altered with aging [12], relationships between aging and intracardiac flow and energetics remain unclear. We hypothesize that aging-related changes in diastolic function may alter intracardiac vortex patterns and reduce energy efficiency. The present study aimed to investigate this hypothesis using VFM.

## 2. Materials and Methods

### 2.1. Study Population

Healthy volunteers were prospectively studied between January 2019 and June 2019. Inclusion criteria were age ≥ 18 years, absence of history of cardiovascular and other chronic diseases, absence of arrhythmia, and normal transthoracic echocardiography. Exclusion criteria were inadequate echocardiographic images or abnormal echocardiographic findings. The ethics committee of our institution approved the study protocol. Written informed consent was obtained from all subjects. This study was registered in the Chinese Clinical Trial Registry (ChiCTR1900022476).

### 2.2. Transthoracic Echocardiography

Two-dimensional and Doppler transthoracic echocardiography was performed in all subjects using the Lisendo 880 ultrasound machine (Hitachi Aloka Medical, Tokyo, Japan). The LV end-diastolic (LVEDV) and end-systolic volume (LVESV), and ejection fraction (LVEF) were measured using the biplane Simpson’s method [13]. The early (E) and late (A) diastolic mitral inflow velocities, the early (e’) and late (a’) diastolic and systolic (s’) medial mitral annular velocities were measured according to the recommendations by the American Society of Echocardiography [14].

### 2.3. Vector Flow Mapping Analysis

Color Doppler acquisitions were performed in the apical long-axis view to allow VFM analysis, with the Nyquist limit increased to minimize aliasing phenomenon, while maintaining a sufficient region of interest to include the entire LV and a frame rate of ≥20 per second. The data were transferred to a DAS-RS1 (Hitachi, Japan) workstation for VFM analysis. The LV endocardium in the end-diastolic frame was traced manually. The software then automatically traced the endocardium throughout the cardiac cycle for speckle-tracking detection of LV wall motions, with manual editing if necessary. Automated anti-aliasing was performed with the same software.

VFM was used to evaluate intracardiac blood flow and viscous EL. Circulation was calculated by using the following equation, equivalent to the integral of normal component of vorticity (*ω*) on an arbitrary plane (***S***) enclosed by a closed curve:$$Circulation=\iint_{S}^{}\omega_{n}dS$$

The areas and circulations of the largest clockwise vortex during systole (S-vortex), early diastole at the time of E wave (E-vortex), and late diastole at the time of A wave (A-vortex) were recorded.

By using the continuity equation, the velocity vector of individual pixels was calculated from information of the adjacent boundaries. A weight function was used to integrate the calculated vectors [15,16]. *EL* was calculated by the following equation:$$EL=\sum i,j\int \frac{1}{2}\mu {\left(\frac{\partial ui}{\partial xj}+\frac{\partial uj}{\partial xi}\right)}^{2}dv$$
where *µ* is the viscosity coefficient of blood, and *ui* and *uj* are the velocity components in the *x* and *y* directions [17]. The peak EL during systole (peak EL_S_), early diastole (peak EL_E_), and late (peak EL_A_) diastole, as well as the mean EL during systole (mean EL_S_) and diastole (mean EL_D_), were measured.

### 2.4. Statistical Analysis

Data were reported as mean ± standard deviation or counts (percentage) when appropriate. The cohort was divided into three groups according to age (18–29 years old, 30–49 years old, and 50 years old or above). Binary variables were compared between age groups using the Chi-square test. The Shapiro–Wilk test was used to test for normality. The Kruskal–Wallis test was used to compare non-parametric continuous variables between the three age groups, while analysis of variance was used to compare parametric continuous variables. Spearman’s correlations evaluated the relationship between VFM profiles and age. All *p*-values were two-sided, with *p* < 0.05 considered statistically significant. All statistical analyses were performed using SPSS software version 25.0 (IBM Corp, New York, NY, USA).

## 3. Results

A total of 100 healthy subjects (33 men; age, 42.9 ± 14.9 years; range, 18–67 years) were included. Demographics, as well as conventional 2D, Doppler, and tissue Doppler echocardiographic findings are summarized in Table 1. There were no differences in LVEDD/ESD, LVEDV/ESV, LVEF, and s’ between the three age groups. Older patients had a significantly lower E/A ratio (*p* < 0.0001) and e’/a’ ratio (*p* < 0.0001), but a higher E/e’ ratio (*p* < 0.0001).

### 3.1. Normal Intracardiac Vortex Flow and EL

In all subjects regardless of age, two intracardiac vortices form during early diastole at the time of the E wave near the LV base, including a larger clockwise vortex (E-vortex) in front of the anterior mitral valve leaflet and a smaller anti-clockwise one behind the posterior mitral valve leaflet. The two vortices grow and migrate towards mid-LV, then fade at the end of the E wave. At the time of the A wave, two vortices form again near the LV base, with a larger anterior clockwise vortex (A-vortex) and a smaller posterior anti-clockwise vortex. The anterior clockwise vortex persists into systole and migrates to the base of heart near the LV outflow tract, reaching its maximal size in mid-systole (S-vortex). A representative example of a normal intracardiac vortex formation is shown in Figure 1 and Appendix A.

The normal LV EL is characterized by three distinct peaks in EL in systole, early diastole, and late diastole, corresponding to the formation of respective intracardiac vortices (Figure 2).

### 3.2. Intracardiac Vortex Flow Characteristics and Aging

Table 2 and Figure 3 compare the intracardiac vortex flow characteristics and EL between younger and older age groups. There were significant age-dependent differences in intracardiac flow, with older patients showing a significantly larger A-vortex area (*p* = 0.0003) with stronger circulation (*p* < 0.0001). The mean EL_S_ was significantly lower than the mean EL_D_ (Wilcoxon signed rank test, *p* < 0.0001) in healthy subjects across all age groups. Older individuals showed a significantly lower peak EL_E_ (*p* = 0.005) but a higher peak EL_A_ (*p* < 0.0001); however, the mean EL_D_ was not significantly different between age groups (*p* = 0.772). There was no intergroup difference in the mean and peak EL_S_.

### 3.3. Interrelations among Intracardiac Flow, Energetics, and Aging

There was a significant correlation between aging and stronger A-vortex circulation (Spearman’s rho (r_s_) = −0.623, *p* < 0.0001). Aging also correlated weakly with weaker E-vortex circulation (r_s_ = 0.197, *p* = 0.049) and stronger S-vortex circulation (r_s_ = −0.218, *p* = 0.029); the correlation between the S-vortex area and aging was not statistically significant (r_s_ = −0.193, *p* = 0.054). Generally, a larger vortex area correlated to stronger circulation, as observed for the S-vortex (r_s_ = −0.463, *p* < 0.0001), E-vortex (r_s_ = −0.386, *p* < 0.0001), and A-vortex (r_s_ = −0.577, *p* < 0.0001).

Peak EL_E_ correlated with a higher mitral inflow E velocity (r_s_ = 0.621, *p* < 0.0001), while peak EL_A_ correlated with a higher mitral inflow A velocity (r_s_ = 0.678, *p* < 0.0001). E-vortex circulation correlated directly to e’ (r_s_ = −0.297, *p* = 0.003), A-vortex circulation correlated directly to A (r_s_ = −0.606, *p* < 0.0001) and a’ (r_s_ = 503, *p* < 0.0001), and S-vortex circulation correlated directly to s’ (r_s_ = −0.214, *p* = 0.032). Additionally, stronger E-vortex circulation correlated to a higher peak EL_E_ (r_s_ = −0.324, *p* = 0.001), while stronger A-vortex circulation correlated to a higher peak EL_A_ (r_s_ = −0.743, *p* < 0.0001). However, there was no significant correlation between S-vortex circulation and the mean and peak EL_S_ (*p* = 0.385 and *p* = 0.374, respectively).

## 4. Discussion

In this study, we confirmed previous descriptions of the pattern of vortex formation and energy loss in healthy individuals [5,18,19,20]. Our study demonstrated important age-related changes in vortex formation and energetics.

Muñoz et al. published a detailed account of the behavior, evolution, and transitions of the intracardiac vortex, which included a detailed analysis of the vortex area and circulation [19]. Our study, with a larger number of subjects, confirmed their findings. It was apparent that the clockwise vortex during diastole, located at the anterior mitral valve leaflet, was larger and persisted longer than the counterclockwise vortex located at the posterior mitral valve leaflet during both passive and active LV filling. Physiologically, such dominance by the clockwise vortex is sensible given the vortex’s role in redirecting blood towards the left ventricular outflow tract. For such reasons, both Muñoz et al. [19] and the current study focused on the clockwise vortex; whether alterations of the anticlockwise vortex bear any physiological or pathological significance remains to be investigated.

Age-related decline in myocardial relaxation and, therefore, LV suction cause an increase in reliance on active atrial filling, with a larger and more intense late diastolic vortex. This differs from patterns observed in young healthy hearts, where active late LV filling generates small vortices that persist into systole [10,20,21,22]. Since the A-vortex persists into systole, our finding of increased A-vortex area and strength with aging is consistent with previous reports in which the mitral inflow A wave was shown to be the only independent predictor of pre-ejection flow velocity in the LV outflow tract [23]. We also found significant correlations between vortex circulations and tissue doppler indices, which reflect that stronger mitral annular motions during LV filling and the ensuing stronger transmitral flow likely result in a stronger vortical flow in the LV. Overall, our results added further mechanistic details for the altered LV filling patterns in aging hearts.

We observed that EL is significantly higher in diastole than in systole, consistent with previous reports [21,24]. This was possibly related to the fact that LV inflow needs to make a “U-turn” at the mid-ventricle and apex. A higher LV filling velocity causes more turbulence colliding with the blood in the LV, causing more EL [25,26]. In contrast, blood flow in the normal LV outflow tract is unidirectionally forward and laminar. Here, intracardiac EL arises from friction between blood flow and the LVOT wall, and the lack of flow collision limits intracardiac EL. Given the preserved systolic function and laminarity of LVOT flow in older individuals, it is unsurprising that intracardiac EL_S_ is preserved. On the other hand, EL_D_ not being elevated in older individuals, despite the changes in diastolic vortex patterns, suggests that the increase in A-vortex area and circulation may be seen as a compensatory response to the decline in E-vortex area and circulation.

Most previous studies of intracardiac flow have used other techniques, such as cardiac magnetic resonance and particle image velocimetry. Though extensively studied, these techniques each have significant limitations: cardiac magnetic resonance is expensive, requires significant time for processing, which prohibits flow visualization at the time of imaging, and is limited in availability at many centers; meanwhile, particle image velocimetry, though allowing for rapid flow visualization at the time of imaging, requires contrast agent injection, often turning an otherwise non-invasive transthoracic echocardiography into an invasive procedure [1]. These may pose difficulties for researchers in recruiting research subjects, and more importantly, they create major barriers for incorporating intracardiac flow imaging into clinical practice in the future. In contrast, VFM is inexpensive, completely non-invasive, and allows for rapid flow visualization at the time of imaging. Future studies may therefore utilize VFM to investigate intracardiac flow in larger cohorts with wider age ranges and possibly follow-up studies, potentially better delineating the intracardiac flow alterations in aging and other conditions, as well as any prognostic significance of these alterations.

### 4.1. Clinical Significance

The close links between diastolic markers, vortex area, and vortex circulation suggest that vortex patterns and intracardiac EL may be deranged in patients with diastolic dysfunction. Data of vortex patterns and intracardiac energetics are scarce for this group of patients [27] and further studies may be worthwhile.

### 4.2. Limitations

Intracardiac vortex is a three-dimensional structure that moves in multiple planes. Visualizing a vortex with a two-dimensional imaging technique such as VFM inherently runs the risk of missing smaller vortices due to the plane of imaging, such as the small vortices that result from the disintegration of the PMVL vortex in late diastole [20]. This problem is inherent to all existing echocardiographic methods of intracardiac flow imaging; three-dimensional VFM is currently being developed, which may be able to overcome this limitation in the near future [28]. Additionally, the analyzed cine loops had minimal frame rates of only at least 20 Hz, which may lead to the underestimation of the largest vortex area and the corresponding circulation, as well as missing any short-lived vortices. The low frame rates may also lead to misinterpretation of aliasing phenomena as EL, which we attempted to minimize by both increasing the Nyquist limit and using the software’s automated anti-aliasing function; such an approach to minimize aliasing has been deployed by previous studies as well [26,29]. Future studies of intracardiac flow imaging should try to achieve higher frame rates, which would allow for a more detailed and accurate delineation of vortex behavior and dimensions throughout the cardiac cycle, as well as preventing inaccuracies in EL measurements due to aliasing. Furthermore, it is unclear if there are significant cycle-to-cycle variations in VFM parameters. Future studies should look to investigate and address this potential issue. Lastly, with the oldest subject being 67 years old, our results’ generalizability to older subjects is uncertain. Nonetheless, finding healthy volunteers at even older ages is difficult and our results remain a solid basis for the understanding of aging-related changes in intracardiac flow.

## 5. Conclusions

In aging hearts where there is an increased dependence on atria-driven LV filling, the early diastolic E-vortex declines in strength, while the late diastolic A-vortex becomes physiologically dominant with increased size and strength. These inverse changes in the E-vortex and A-vortex were accompanied by maintained energetic efficiency. Further investigations of vortex imaging may yield clinically significant findings in a variety of conditions.

## Figures and Tables

**Figure 1 jcm-10-03619-f001:**
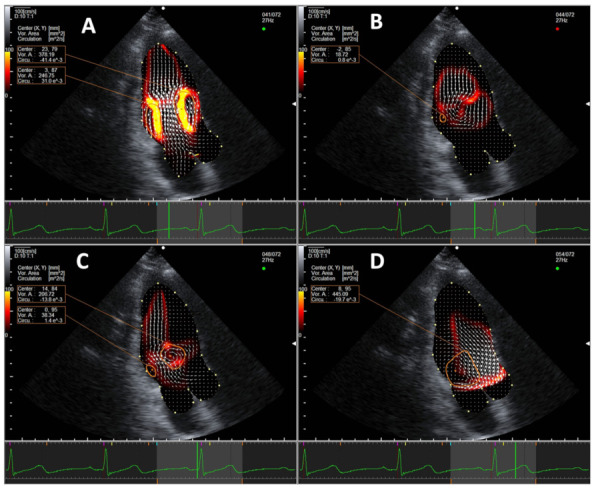
Representative example of normal vortex formation pattern in the apical three-chamber view during (**A**) early diastole, (**B**) at the completion of passive ventricular filling, (**C**) during late diastole, and (**D**) during systole. Vortices are marked in orange circles. An energy loss heat map is displayed under the velocity vectors (white arrows), with red being low energy loss and yellow being high.

**Figure 2 jcm-10-03619-f002:**
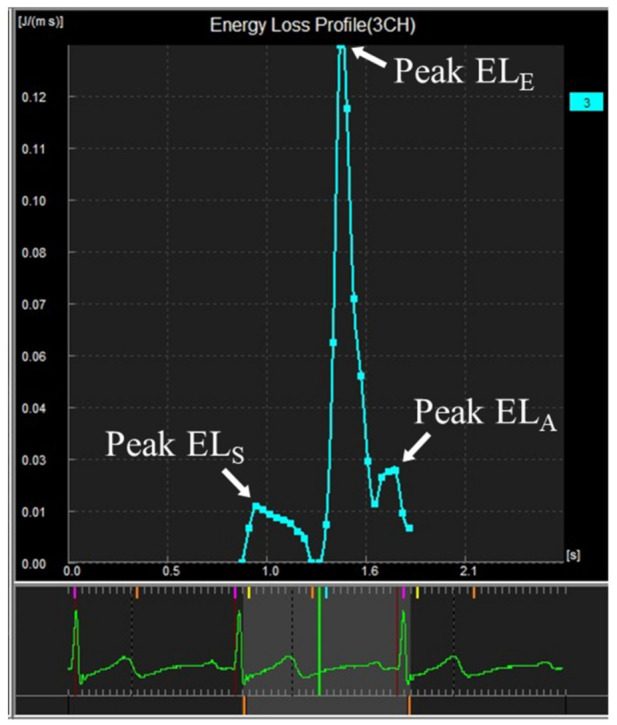
Curve showing intracardiac energy loss by VFM analysis in a young subject.

**Figure 3 jcm-10-03619-f003:**
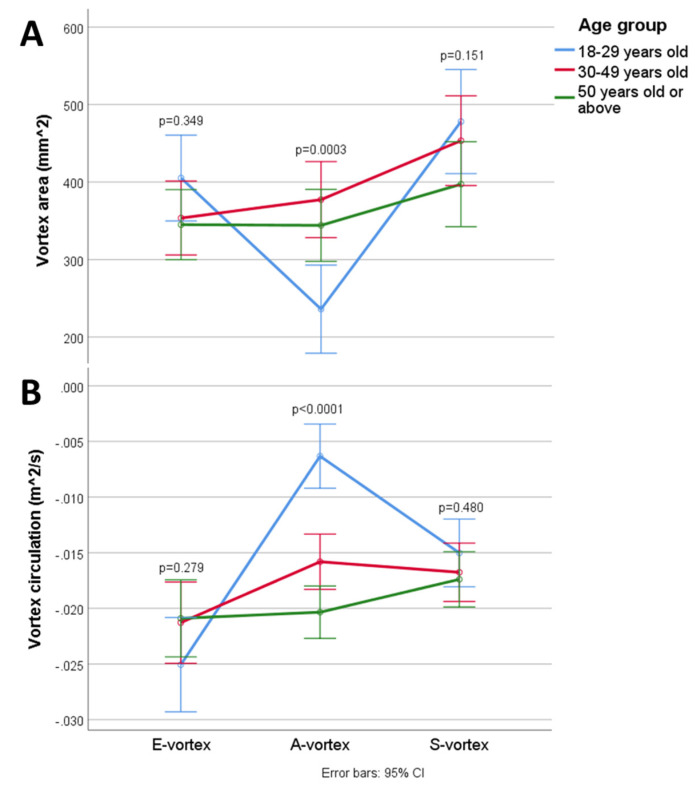
Changes in the (**A**) vortex area and (**B**) circulation in the cardiac cycle.

**Table 1 jcm-10-03619-t001:** Demographics and two-dimensional and Doppler echocardiographic parameters stratified by age groups.

	Age Groups			
	18−29 Years (*n* = 26)	30–49 Years (*n* = 35)	50 Years or Above (*n* = 39)	*p* Value
Age, years	22.4 ± 2.5	41.3 ± 5.8	58.0 ± 4.9	<0.0001
BSA, m^2^	1.68 ± 0.19	1.75 ± 0.25	1.63 ± 0.16	0.097
Female, N (%)	16 (61.5)	23 (65.7)	28 (71.8)	0.782
LVEDD, mm	44.1 ± 4.7	44.1 ± 4.6	43.4 ± 25.1	0.886
LVESD, mm	26.5 ± 4.1	26.9 ± 3.6	25.1 ± 3.7	0.102
LVEDV, mL	78.8 ± 17.1	75.7 ± 19.1	24.5 ± 5.8	0.117
LVESV, mL	26.2 ± 5.3	25.6 ± 6.9	65.4 ± 2.6	0.489
LVEF, %	66.6 ± 2.43	66.3 ± 2.3	71.7 ± 14.7	0.139
E, cm/s	84.0 ± 19.4	75.0 ± 14.5	60.6 ± 14.2	0.011
A, cm/s	34.9 ± 13.6	53.2 ± 12.4	60.6 ± 14.2	<0.0001
E/A	2.70 ± 1.11	1.48 ± 0.40	1.25 ± 0.42	<0.0001
e’, cm/s	12.3 ± 0.80	9.64 ± 2.07	8.68 ± 1.56	<0.0001
a’, cm/s	8.28 ±1.58	9.54 ± 1.84	10.6 ± 1.96	<0.0001
s’, cm/s	7.90 ± 1.06	7.75 ± 1.14	7.77 ± 1.62	0.905
e’/a’	1.52 ± 0.33	1.05 ± 0.30	0.845 ± 0.216	<0.0001
E/e’	6.99 ±2.01	8.02 ± 1.91	8.40 ± 1.82	<0.0001

BSA, body surface area. LVEDD, left ventricular end-diastolic diameter; LVEDV, left ventricular end-diastolic volume; LVESD, left ventricular end-systolic diameter; LVESV, left ventricular end-systolic volume; LVEF, left ventricular ejection fraction; E, early diastolic mitral inflow velocity; A, late diastolic mitral inflow velocity; e’, early diastolic mitral annulus velocity; a’, late diastolic mitral annulus velocity; s’, systolic mitral annulus velocity.

**Table 2 jcm-10-03619-t002:** Comparison of vector flow mapping results by age groups.

	Age Groups			
	18−29 Years (*n* = 26)	30–49 Years (*n* = 35)	50 Years or Above (*n* = 39)	*p* Value
E-Vortex area, mm^2^	405 ± 169	354 ± 124	345 ± 138	0.349
E-Vortex circulation, m^2^/s	−0.0251 ± 0.0149	−0.0213 ± 0.0085	−0.0209 ± 0.0097	0.279
A-Vortex area, mm^2^	236 ± 160	377 ± 141	344 ± 141	0.0003
A-Vortex circulation, m^2^/s	−0.00632 ± 0.00409	−0.0158 ± 0.0073	−0.0203 ± 0.0090	<0.0001
S-Vortex area, mm^2^	478 ± 172	453 ± 172	397 ± 174	0.151
S-Vortex circulation, m^2^/s	−0.0150 ± 0.0079	−0.0168 ± 0.0078	−0.0174 ± 0.0078	0.480
Peak EL_E_, J/m^3^s	35.5 ± 17.9	24.3 ± 15.8	22.2 ± 13.7	0.005
Peak EL_A_, J/m^3^s	3.42 ± 2.83	8.03 ± 5.71	12.3 ± 8.6	<0.0001
Peak EL_S_, J/m^3^s	10.6 ± 7.2	8.54 ± 4.48	10.4 ± 6.67	0.714
Mean EL_D_, J/m^3^s	9.59 ± 5.52	8.96 ± 5.27	9.93 ± 5.85	0.772
Mean EL_S_, J/m^3^s	4.27 ± 2.60	4.30 ± 2.21	4.93 ± 2.68	0.297

E, early diastole; A, late diastole; S, systole; EL, energy loss.

## Data Availability

The data presented in this study are available on request from the corresponding author. The data are not publicly available due to privacy reasons and restrictions by the protocol.

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
