# Peer review of "Age-Related Changes in Left Ventricular Vortex Formation and Flow Energetics"

_jcm, 2021, doi:10.3390/jcm10163619_

Round 1
Reviewer 1 Report
The study of Dr Chan et al takes a step in an uncharted terrain of intracardiac flow dynamics. The terrain remains uncharted despite the already over 10 years of describing the method, because there is doubt concerning the real clinical significance and the way this information can be used clinically. Even the 4D flow MRI, being 3D and producing beautiful images, in the end does not give clinical insights that were impossible to obtain with other less visually striking methods. That being said, we will only advance if we search further, and this is the merit of this paper. However, I do have some concerns regarding the interpretation of the results:
- The main problem is not the way the study is made, but the way the data is interpreted. Data is good and valuable. But you should state only what is derived from your results, not overstate or use speculations as proven data. In this category enter: stating the good feasibility of the method (see explanation below), stating the robustness of the method, the reliability of the method, without really investigating them properly (see below).
- You should discuss the real limitations of the method (see comments below) and of the intracardiac flow tracking in general (no clear clinical implication in general).
Other comments as they arise during the lecture of the manuscript, and clarifications to the main comments:
Abstract
- The obvious relation of the vortex parameters with E, A, e’ and s’ is ignored. See my later comments
Introduction
- I miss a movie with the whole cardiac cycle
Methods
- Given the number of variables and the number of age categories, I fear that the real age evolution of the vortex patterns would require a larger population to clearly ascertain. But nevertheless this remains valuable.
- Frame rate was a little over 20 Hz!
- Ideally the division of groups should have included more categories, maybe every 2 decades. I do not know exactly how this would have changed your results and groups distribution, but I would love to see the time evolution per decades for example, maybe there is a trend visible.
- Age repartition seems to me insufficient for real assessment of aging of the heart. After all max age was 67, while life expectancy currently revolves around 80 in China.
Results
- I would expect that VFM be feasible in all patients, this was a selection criterion (good image quality and adequate frame rate). This total feasibility sounds good but is scientifically not correct.
- The description of the vortex evolution between text and the Figure 1 legend is repetitive. Please adjust to make it rather complementary.
- I am wondering when I look at the images and at the EL graphic whether the energy loss is really loss or rather energy transmission/dissipation. After all this energy is basically kinetic energy, because it is computed from velocity fields.
- The result from Figure 2 is in young or older patients?
- I would like to see the age repartition of your groups. It is totally different to have for example 30 patients of around 20 y-o in the first group and 40 patients over 60 in the second, than to have a more homogeneous distribution.
- I am not sure that you may call “intracardiac energetic performance”, because the real energetic performance of the heart is something totally different (equally difficult to define). I would rather suggest intracardiac flow energy dissipation/transmission/or even loss.
- An r value of 0.616 depicts a moderate correlation, which is statistically significant. I am not entirely sure that the baseline requirements and assumptions for a Pearson’s test are observed (I would like to know if the variable distribution was checked for normality).
- Age or aging correlated with E-vortex circulation? This correlation was weak (r=0.215)
- It is normal that peak EL would correlate with flow velocity. The EL is derived from a velocity equation. Also, correlation with circulation are statistically disputable, because circulation represents somewhat all the velocities in the vortex and EL somehow the main direction velocity vectors which “push” energy into the vortex. Because energy tends to be stable, the inflow kinetic energy, the elastic wall distension and recoil, contraction and outflow should sum up to zero.
Inter/intra observer
I am not entirely sure of the necessity for intra/inter observer analysis. The way you presented the Methods I have the tendency to believe that much of the calculation is automatic, based on a software analysis within a ROI dynamically adapted to the wall motion. This is expected to provide the same results in the same ROI. If I am wrong, please present the subjective steps that are sensitive to individual variation. I suggest that you describe the way you obtain the flow/energy parameters and also illustrate them with a whole heart cycle movie.
Discussion
- You did not state in your aims that you wanted to provide a feasibility and reproducibility study. While the reproducibility is analyzed (as pointed above, I did not understand why this analysis was necessary, please correct me if I am wrong), feasibility is not, as pointed out in my earlier remark regarding patient selection.
- You talk about the clockwise diastolic flow, but there are two vortices or two stages of a vortex, depending on the length of the diastasis. The E vortex would certainly not contribute to the LVOT flow.
- We do not know whether the vortices in young adults facilitate ejection, it is speculative.
- It is very true that the findings of this study merely confirm the evolution of the E/A pattern with age.
- EL in systole from the perspective of the apical long axis does not analyze the true place where systolic EL takes place, this means in the ascending aorta. There is little EL because of laminar flow in normal LVOT, vortices will form after the aortic annulus. This is more likely than the U-turn of diastolic inflow. The energy of the systolic flow is clearly higher, taking in much of the inflow energy, the elastic recoil of the walls and the contraction.
- I am not entirely convinced that the vortices explain the hemodynamic and energetic basis of the atrial work.
Clinical significance
- As I read your introduction I have the impression that the method is already mature enough, since published data comes from 2016 already, and a clinical application is already present on the Hitachi systems. I do not see why we need this reproducibility study which was not in the aim of the paper. As for being reliable, this is not demonstrated in this study, because you do not have any reference method.
- Also, your results do not confirm the role of A vortex in aiding ejection, you only speculate on the timing of A vortex and assume it may aid with the ejection flow. We are also not talking about the overall energetic efficiency of the heart, but to flow dynamics energy efficiency. The heart spends a lot more energy on contraction and relaxation, and its translation in flow velocity and pressure depends of numerous other factors, like tissue elasticity, geometry, postcharge, and of course flow dynamics.
- You state that vortex assessment may be important in AF patients, based on A vortex role. But of course, A vortex would be absent in AF, so which vortex are you planning to analyze?
- The close link between diastolic markers (velocities) and vortex characteristics (velocity derived) are partially due to the equations which serve to compute the vortex parameters. They will most probably be modified in diastolic dysfunction, because the inflow velocities drive these vortices. But I do agree that data is scarce and further studies should be performed.
Limitations
- The correlations you present are often moderate or weak, and may be due to an effect-cause mechanism, so they do not sufficiently prove the method is robust.
- Another limitation of the method is that is derived from Doppler, so it is angle-dependent. Vortex motion that is perpendicular to the ultrasound direction may be poorly or not at all visualized.
- Another limitation is the frame rate (somewhere 20-30 Hz, not presented). Higher velocities would be wrongly represented, and some energy loss may be in fact misinterpretation of aliasing phenomena. High frame rate echocardiography has already delivered very interesting results using echoPIV, and measuring the real velocity of the intracardiac flow. See Voorneveld J et al. High-Frame-Rate Echo-Particle Image Velocimetry Can Measure the High-Velocity Diastolic Flow Patterns. Circ Cardiovasc Imaging. 2019;12(4):e008856. This should be discussed, because we do not yet know which method of flow tracking would finally be used clinically. And as also presented in your ref. 2 Mele et al 2018, we do not exactly know what results from all these vortex parameters, and after the beautiful images and complex equations, the clinical cardiologist still can say ”so what?...”
Conclusion: correctly formulated, state the findings as they are.
Supplemental material: maybe it is my computer, but the video is not one flowing complete heart cycle, but a sequence of still frames. I personally would like to see a complete heart cycle, be that with less explaining and indications.
Author Response
Comments from Reviewer 1
The study of Dr Chan et al takes a step in an uncharted terrain of intracardiac flow dynamics. The terrain remains uncharted despite the already over 10 years of describing the method, because there is doubt concerning the real clinical significance and the way this information can be used clinically. Even the 4D flow MRI, being 3D and producing beautiful images, in the end does not give clinical insights that were impossible to obtain with other less visually striking methods. That being said, we will only advance if we search further, and this is the merit of this paper. However, I do have some concerns regarding the interpretation of the results:
- The main problem is not the way the study is made, but the way the data is interpreted. Data is good and valuable. But you should state only what is derived from your results, not overstate or use speculations as proven data. In this category enter: stating the good feasibility of the method (see explanation below), stating the robustness of the method, the reliability of the method, without really investigating them properly (see below).
Authors’ response: Thank you for your comment. We have revised according to your comments and have detailed our point-to-point responses below.
- You should discuss the real limitations of the method (see comments below) and of the intracardiac flow tracking in general (no clear clinical implication in general).
Authors’ response: Thank you for your comment. We have revised according to your comments and have detailed our point-to-point responses below.
Other comments as they arise during the lecture of the manuscript, and clarifications to the main comments:
Abstract
- The obvious relation of the vortex parameters with E, A, e’ and s’ is ignored. See my later comments
Authors’ response: Thank you for your comment. We have added the results in lines 161-164: “E-vortex circulation correlated directly to e’ (rs=-0.297, p=0.003), A-vortex circulation correlated directly to A (rs=-0.606, p<0.0001) and a’ (rs=503, p<0.0001), and S-vortex circulation correlated directly to s’ (rs=-0.214, p=0.032).”
Introduction
- I miss a movie with the whole cardiac cycle
Authors’ response: Thank you for your comment. Since all VFM movies were obtained as a result of our study, we have decided to keep the movie within the Results section instead.
Methods
- Given the number of variables and the number of age categories, I fear that the real age evolution of the vortex patterns would require a larger population to clearly ascertain. But nevertheless this remains valuable.
Authors’ response: Thank you for your comment. We agree that a larger cohort is needed to delineate more clearly the effects of aging on intracardiac flow. We have added a relevant statement in the Discussion addressing this (lines 215-218): “Future studies may therefore utilize VFM to investigate intracardiac flow in larger cohorts with wider age ranges and possibly follow-up studies, potentially better delineate the intracardiac flow alterations in aging and other conditions”.
- Frame rate was a little over 20 Hz!
Authors’ response: Thank you for your comment. This was unfortunately limited by the system we used. We have added this as a limitation (lines 232-236): “Additionally, the analysed cine loops had frame rates of only at least 20 Hz, which may lead to underestimation of the largest vortex area and the corresponding circulation. The low frame rates may also lead to misinterpretation of aliasing phenomena as EL, which we attempted to minimize by using the software’s automated anti-aliasing function.”
- Ideally the division of groups should have included more categories, maybe every 2 decades. I do not know exactly how this would have changed your results and groups distribution, but I would love to see the time evolution per decades for example, maybe there is a trend visible.
Authors’ response: Thank you for your comment. In the manuscript, 45 years old was chosen as the cut-off since it was the median. We have now changed to using 3 age groups instead.
- Age repartition seems to me insufficient for real assessment of aging of the heart. After all max age was 67, while life expectancy currently revolves around 80 in China.
Authors’ response: Thank you for your comment. While we agree that our study may not reflect the full spectrum of aging due to the age range of included patients, it is exceptionally difficult to find healthy volunteers without any known history of cardiovascular or chronic diseases beyond the age of 65. While we acknowledge that the current results’ may not be directly generalizable to even older individuals, we are of the opinion that our results still provide a solid basis for the understanding of changes in intracardiac flow in normal aging, and that it is unlikely to have healthy volunteers at ages of 70s and 80s for similar characterizations. We have nonetheless acknowledged this potential limitation as the last point of our Limitations section (lines 236-239): “Lastly, with the oldest subject being 67 years old, our results’ generalizability to older subjects is uncertain. Nonetheless, finding healthy volunteers at even older ages is difficult and our results remain a solid basis for the understanding of aging-related changes in intracardiac flow.”
Results
- I would expect that VFM be feasible in all patients, this was a selection criterion (good image quality and adequate frame rate). This total feasibility sounds good but is scientifically not correct.
Authors’ response: Thank you for your comment. We agree that the feasibility statement is redundant and misleading and have therefore removed the statement.
- The description of the vortex evolution between text and the Figure 1 legend is repetitive. Please adjust to make it rather complementary.
Authors’ response: Thank you for your comment. We have simplified the caption of Figure 1. It now reads: “Representative example of normal vortex formation pattern in the apical 3-chamber view during early diastole (top left), at the completion of passive ventricular filling (top right), during late diastole (bottom left), and during systole (bottom right). Vortices are marked in orange circles. Energy loss heat map is displayed under the velocity vectors (white arrows), with red being low energy loss and yellow being high.”
- I am wondering when I look at the images and at the EL graphic whether the energy loss is really loss or rather energy transmission/dissipation. After all this energy is basically kinetic energy, because it is computed from velocity fields.
Authors’ response: Thank you for your comment. While we agree that EL may be better termed as energy dissipation, the term “energy loss” has been used consistently in previous publications of VFM and intracardiac flow analysis. To better frame this concept, we have edited the statement in the introduction that explained what EL is (lines 35-36): “… intracardiac haemodynamics, including energy loss (EL) which is the energy dissipated via frictional heat generated due to viscosity of blood at sites of turbulent flow.”
- The result from Figure 2 is in young or older patients?
Authors’ response: Thank you for your comment. It was from a young subject. We have edited the Figure 2 caption accordingly. It now reads: “Curve showing intracardiac energy loss by VFM analysis in a young subject.”
- I would like to see the age repartition of your groups. It is totally different to have for example 30 patients of around 20 y-o in the first group and 40 patients over 60 in the second, than to have a more homogeneous distribution.
Authors’ response: Thank you for your comment. We have changed the grouping to 3 groups as per your suggestion.
- I am not sure that you may call “intracardiac energetic performance”, because the real energetic performance of the heart is something totally different (equally difficult to define). I would rather suggest intracardiac flow energy dissipation/transmission/or even loss.
Authors’ response: Thank you for your comment. We have changed the wording to simply “energy loss (EL)” and avoided comments on “energetic performance” throughout the manuscript.
- An r value of 0.616 depicts a moderate correlation, which is statistically significant. I am not entirely sure that the baseline requirements and assumptions for a Pearson’s test are observed (I would like to know if the variable distribution was checked for normality).
Authors’ response: Thank you for your comment. We have performed the Shapiro-Wilk test for normality as part of this Revision and have revised the Statistical analysis section accordingly. After reviewing the data, we have decided to use Spearman’s correlation instead of Pearson’s correlation. The relevant section now reads (lines 96-99): “The Shapiro-Wilk test was used to test for normality. The Kruskal-Wallis test was used to compare non-parametric continuous variables between the three age groups, while analysis of variance was used to compare parametric continuous variables. Spearman’s correlations evaluated the relationship between VFM profiles and age.” Additionally, we have refrained from labelling correlations as “strong”.
- Age or aging correlated with E-vortex circulation? This correlation was weak (r=0.215)
Authors’ response: Thank you for your comment. We have acknowledged this by adding the appropriate label in line 154: “Aging also correlated weakly with weaker E-vortex circulation (rs=0.197, p=0.049) and stronger S-vortex circulation (rs=-0.218, p=0.029)”.
- It is normal that peak EL would correlate with flow velocity. The EL is derived from a velocity equation. Also, correlation with circulation are statistically disputable, because circulation represents somewhat all the velocities in the vortex and EL somehow the main direction velocity vectors which “push” energy into the vortex. Because energy tends to be stable, the inflow kinetic energy, the elastic wall distension and recoil, contraction and outflow should sum up to zero.
Authors’ response: Thank you for your comment. We appreciate that there are common components between EL and circulation. Nonetheless, the end products (EL and circulation) represent rather different entities, and the varying degrees of correlations remain potentially informative. In addition, we found no correlation between systolic EL and S-vortex circulation, suggesting that additional factors may be at play.
Inter/intra observer
I am not entirely sure of the necessity for intra/inter observer analysis. The way you presented the Methods I have the tendency to believe that much of the calculation is automatic, based on a software analysis within a ROI dynamically adapted to the wall motion. This is expected to provide the same results in the same ROI. If I am wrong, please present the subjective steps that are sensitive to individual variation. I suggest that you describe the way you obtain the flow/energy parameters and also illustrate them with a whole heart cycle movie.
Authors’ response: Thank you for your comment. Indeed, most of the steps of the analysis were automated. We have therefore removed the inter/intraobserver reliability section.
Discussion
- You did not state in your aims that you wanted to provide a feasibility and reproducibility study. While the reproducibility is analyzed (as pointed above, I did not understand why this analysis was necessary, please correct me if I am wrong), feasibility is not, as pointed out in my earlier remark regarding patient selection.
Authors’ response: Thank you for your comment. We have removed the statement about feasibility and reliability accordingly.
- You talk about the clockwise diastolic flow, but there are two vortices or two stages of a vortex, depending on the length of the diastasis. The E vortex would certainly not contribute to the LVOT flow.
Authors’ response: Thank you for your comment. We apologize for being unclear – the statement was referring to both the E- and A-vortex. We have edited the statement (line 178) to improve clarity; the statement now reads: “It was apparent that the clockwise vortex during diastole, located at the anterior mitral valve leaflet, was larger and persisted longer than the counterclockwise vortex located at the posterior mitral valve leaflet during both passive and active LV filling.”
- We do not know whether the vortices in young adults facilitate ejection, it is speculative.
Authors’ response: Thank you for your comment. We have removed the relevant statements accordingly.
- It is very true that the findings of this study merely confirm the evolution of the E/A pattern with age.
Authors’ response: Thank you for your comment. This is understood.
- EL in systole from the perspective of the apical long axis does not analyze the true place where systolic EL takes place, this means in the ascending aorta. There is little EL because of laminar flow in normal LVOT, vortices will form after the aortic annulus. This is more likely than the U-turn of diastolic inflow. The energy of the systolic flow is clearly higher, taking in much of the inflow energy, the elastic recoil of the walls and the contraction.
Authors’ response: Thank you for your comment. We appreciate that the systolic EL in terms of the cardiovascular system arises mainly from turbulence in the ascending aorta. Our elaboration in the Discussion, as corroborated by your comment above, explains the lower intracardiac systolic EL only. Systolic EL in the ascending aorta involves complex vortex interactions that warrant an entire, separate study. It is out of the scope of the current study, and we have therefore decided to not discuss the systolic EL in the ascending aorta. Nonetheless, to ensure clarity, we have specifically added the label “intracardiac EL” when discussing the systolic EL in the Discussion (lines 213, 215, 216): “Here, intracardiac EL arises from friction between blood flow and the LVOT wall, and the lack of flow collision limits intracardiac EL. Given the preserved systolic function and laminarity of LVOT flow in older individuals, it is unsurprising that intracardiac ELS is preserved.”
- I am not entirely convinced that the vortices explain the hemodynamic and energetic basis of the atrial work.
Authors’ response: Thank you for your comment. We have removed that statement accordingly.
Clinical significance
- As I read your introduction I have the impression that the method is already mature enough, since published data comes from 2016 already, and a clinical application is already present on the Hitachi systems. I do not see why we need this reproducibility study which was not in the aim of the paper. As for being reliable, this is not demonstrated in this study, because you do not have any reference method.
Authors’ response: Thank you for your comment. We have removed the entire section about reproducibility, as well as all statements concerning reliability / feasibility.
- Also, your results do not confirm the role of A vortex in aiding ejection, you only speculate on the timing of A vortex and assume it may aid with the ejection flow. We are also not talking about the overall energetic efficiency of the heart, but to flow dynamics energy efficiency. The heart spends a lot more energy on contraction and relaxation, and its translation in flow velocity and pressure depends of numerous other factors, like tissue elasticity, geometry, postcharge, and of course flow dynamics.
Authors’ response: Thank you for your comment. The contribution of the A vortex to systolic ejection has been shown by Govindarajan et al (see citation 19 in the manuscript). Nonetheless, we agree that our data cannot definitively show this, and it is not the focus of our findings. We have therefore removed the relevant statements.
- You state that vortex assessment may be important in AF patients, based on A vortex role. But of course, A vortex would be absent in AF, so which vortex are you planning to analyze?
Authors’ response: Thank you for your comment. We apologize for the confusion – we meant to suggest that intracardiac flow analysis in general (including EL etc), instead of focusing on just one vortex, might be useful in patients with AF. Nonetheless, since we have removed the parts about A-vortex contributing to systolic ejection, we have removed the captioned paragraph altogether.
- The close link between diastolic markers (velocities) and vortex characteristics (velocity derived) are partially due to the equations which serve to compute the vortex parameters. They will most probably be modified in diastolic dysfunction, because the inflow velocities drive these vortices. But I do agree that data is scarce and further studies should be performed.
Authors’ response: Thank you for your comment. We agree that this is a logical and intuitive extension of our findings.
Limitations
- The correlations you present are often moderate or weak, and may be due to an effect-cause mechanism, so they do not sufficiently prove the method is robust.
Authors’ response: Thank you for your comment. We have removed the relevant comment.
- Another limitation of the method is that is derived from Doppler, so it is angle-dependent. Vortex motion that is perpendicular to the ultrasound direction may be poorly or not at all visualized.
Authors’ response: Thank you for your comment. Although VFM involves data obtained by colour Doppler, it combines such data with measurements from speckle-tracking echocardiography to calculate the direction and velocity of intracardiac blood flow. As such, it is well established that VFM has no angle dependency, and vortex motions perpendicular to the ultrasound direction can be visualized without problem.
- Another limitation is the frame rate (somewhere 20-30 Hz, not presented). Higher velocities would be wrongly represented, and some energy loss may be in fact misinterpretation of aliasing phenomena. High frame rate echocardiography has already delivered very interesting results using echoPIV, and measuring the real velocity of the intracardiac flow. See Voorneveld J et al. High-Frame-Rate Echo-Particle Image Velocimetry Can Measure the High-Velocity Diastolic Flow Patterns. Circ Cardiovasc Imaging. 2019;12(4):e008856. This should be discussed, because we do not yet know which method of flow tracking would finally be used clinically. And as also presented in your ref. 2 Mele et al 2018, we do not exactly know what results from all these vortex parameters, and after the beautiful images and complex equations, the clinical cardiologist still can say ”so what?...”
Authors’ response: Thank you for your comments. We have added low frame rate as one of the limitations. In regard to the issue of misinterpreting aliasing as energy loss, we have attempted to minimize this by using the Hitachi software’s automated anti-aliasing function. We have nonetheless acknowledged this potential limitation in the same section. The revised section now reads (lines 232-236): “Additionally, the analysed cine loops had frame rates of only at least 20 Hz, which may lead to underestimation of the largest vortex area and the corresponding circulation. The low frame rates may also lead to misinterpretation of aliasing phenomena as EL, which we attempted to minimize by using the software’s automated anti-aliasing function.”
Conclusion: correctly formulated, state the findings as they are.
Authors’ response: Thank you for your comments. We have also edited the conclusion to reflect the changes in results as a consequence of the modified statistical analyses approaches.
Supplemental material: maybe it is my computer, but the video is not one flowing complete heart cycle, but a sequence of still frames. I personally would like to see a complete heart cycle, be that with less explaining and indications.
Authors’ response: Thank you for your comments. The supplementary video was consecutive frames from cine loops of one complete cardiac cycle, which only appeared as a sequence of still frames because we slowed it down significantly. At its original speed, the vortex tracings move around too quickly for viewing and therefore we decided to slow it down such that the progression of vortex patterns through a cardiac cycle could be better appreciated.
Reviewer 2 Report
The paper is well written. Article presentation is clear and sufficiently accurate. Results are well in line with the hypothesis. The subject is relatively novel and of interest.
There are some observation to this paper:
-too limited sample size divided by only two broad group of age
-use of VFI instead of high-frame rate VFI
- not very clear inclusion/exclusion criteria and gender differentiation
Author Response
Comments from Reviewer 2
The paper is well written. Article presentation is clear and sufficiently accurate. Results are well in line with the hypothesis. The subject is relatively novel and of interest.
Authors’ response: Thank you for your comments.
There are some observation to this paper:
-too limited sample size divided by only two broad group of age
Authors’ response: Thank you for your comment. To acknowledge this, we have added a statement that explored the utility of larger future trials and follow-up studies (lines 252-256). Additionally, and as per reviewer 1’s comments, we have now switched to a 3-group approach with all results updated accordingly.
-use of VFI instead of high-frame rate VFI
Authors’ response: Thank you for your comment. This was unfortunately limited by the system we used. We have added this as a limitation (lines 232-236): “Additionally, the analysed cine loops had minimal frame rates of only at least 20 Hz, which may lead to underestimation of the largest vortex area and the corresponding circulation, as well as missing any short-lived vortices. The low frame rates may also lead to misinterpretation of aliasing phenomena as EL, which we attempted to minimize by using the software’s automated anti-aliasing function.”
- not very clear inclusion/exclusion criteria and gender differentiation
Authors’ response: Thank you for your comment. We have slightly edited section 2.1. The inclusion and exclusion criteria have been stated very early in the same paragraph as well (lines 53-56). The section now reads: “Healthy volunteers were prospectively studied between January 2019 and June 2019. Inclusion criteria were age ≥18 years, absence of history of cardiovascular and other chronic diseases, absence of arrhythmia, and normal transthoracic echocardiography. Exclusion criteria were inadequate echocardiographic images or abnormal echocardiographic findings.” We hope that this is sufficiently clear.
In addition, we have added the number and percentage of females to Table 1 and compared it across age groups in order to demonstrate the homogeneity of sex distribution across age groups. Since the focus of this study is on aging and not inter-sex differences, we did not study the inter-sex differences in flow parameters.
Reviewer 3 Report
The authors describe age-related changes in left ventricular vortex formation and flow energetics. This subject is of clinical interest and previous research on this topic is limited. The draft is well written, and the video looks nice. However, there are several methodological issues that needs to be better explained and discussed in this work.
- Vector flow mapping analysis: Why did you choose to present “circulation” and not “vorticity” in your measurements?
- You state that the framerate was ≥ 20 per second. This limit is set quite low. Flow events are very short-lived. What was the mean framerate in your data? How can a relatively low framerate influence your measurements (and conclusions)?
- Intra- and inter observer variability analysis: Was this done only for the postprocessing part, or did it also include imaging? Did you measure VFM parameters from the same cardiac cycle?
- Did you investigate if the VFM measurements such as EL and “circulation” varied from cycle to cycle in each patient?
- In my experience vortex sizing could be difficult if unclear borders. How did you define the borders of the vortex in your study? I.e in figure 1, lower left it seems like part of the vortex area (to the right of the vortex) is outside the drawing for the central vortex.
- Results – Demographics: Do you have information regarding the BMI in the two age groups? Do you think a significantly higher BMI in one of the groups would have influenced the results?
- In the last part of the discussion, you state that “vortex circulation and energy loss were relatively independent of imaging plane”. Did you show this in your data? This contrasts with previous studies (both in silico and in vivo) that demonstrates significant changes in EL with imaging plane. Please discuss.
Author Response
Comments from Reviewer 3
The authors describe age-related changes in left ventricular vortex formation and flow energetics. This subject is of clinical interest and previous research on this topic is limited. The draft is well written, and the video looks nice. However, there are several methodological issues that needs to be better explained and discussed in this work.
- Vector flow mapping analysis: Why did you choose to present “circulation” and not “vorticity” in your measurements?
Authors’ response: Thank you for your comment. Quoting the Hitachi system’s description, vorticity refers to the “direction and intensity of the rotating movement of flow velocity vectors at a focal point”, implying that it is a per-point, instead of per-vortex, measurement. There are multiple points of such measurements within a single vortex, none of which is individually representative of the entire vortex’s intensity. In contrast, circulation represents the sum of vorticities within a vortex. Therefore, in order to analyse the intensity of selected vortices, circulation should be used instead of vorticity.
- You state that the framerate was ≥ 20 per second. This limit is set quite low. Flow events are very short-lived. What was the mean framerate in your data? How can a relatively low framerate influence your measurements (and conclusions)?
Authors’ response: Thank you for your comment. Unfortunately, the frame rate was limited by the system used at the time of imaging, and the frame rate was not recorded as part of the data collection process. This concern was also raised by several other reviewers and we have added this as one of the limitations (lines 232-236): “Additionally, the analysed cine loops had minimal frame rates of only at least 20 Hz, which may lead to underestimation of the largest vortex area and the corresponding circulation, as well as missing any short-lived vortices. The low frame rates may also lead to misinterpretation of aliasing phenomena as EL, which we attempted to minimize by using the software’s automated anti-aliasing function.”
- Intra- and inter observer variability analysis: Was this done only for the postprocessing part, or did it also include imaging? Did you measure VFM parameters from the same cardiac cycle?
Authors’ response: Thank you for your comment. We only did this for the postprocessing part with the measurements done on the same cardiac cycle. Nonetheless, we have removed the intra- and interobserver reproducibility analysis as per Reviewer 1’s comments.
- Did you investigate if the VFM measurements such as EL and “circulation” varied from cycle to cycle in each patient?
Authors’ response: Thank you for your comment. Such cycle-to-cycle variations were not evaluated as it was out the scope of the current study. We acknowledge that this is a potential limitation which deserves investigation in the future, and have therefore added it to the limitations section (lines 236-239): “Furthermore, only one cardiac cycle was analysed per patient, and it is unclear if there is any significant cycle-to-cycle variations in VFM parameters. Future studies should look to investigate and address this potential issue.”
- In my experience vortex sizing could be difficult if unclear borders. How did you define the borders of the vortex in your study? I.e in figure 1, lower left it seems like part of the vortex area (to the right of the vortex) is outside the drawing for the central vortex.
Authors’ response: Thank you for your comment. The borders of the vortex were defined automatically by the Hitachi software.
- Results – Demographics: Do you have information regarding the BMI in the two age groups? Do you think a significantly higher BMI in one of the groups would have influenced the results?
Authors’ response: Thank you for your comment. We are of the opinion that for comparison of anthropometric measurements, body surface area (BSA) would be more suitable than BMI as it is more routinely used for adjusting echocardiographic measurements. We have now added BSA to Table 1 and shown that it is not significantly different between age groups (which have been changed to a 3-group system as per Reviewer 1’s comments).
- In the last part of the discussion, you state that “vortex circulation and energy loss were relatively independent of imaging plane”. Did you show this in your data? This contrasts with previous studies (both in silico and in vivo) that demonstrates significant changes in EL with imaging plane. Please discuss.
Authors’ response: Thank you for your comment. We acknowledge that this was not shown by our data and we have removed the statement accordingly. We have also added a line mentioning the active development of 3D VFM (lines 230-232): “3-dimensional VFM is currently being developed, which may be able to overcome this limitation in the near future.”
Reviewer 4 Report
Comments
Authors described the evaluation for the age-related cardiac changes, which were mainly diastolic dysfunction, by using vector flow mapping (VFM) analysis of intracardial flow.
Authors also calculated peak energy loss (EL) for comparison between young and aging hearts. Decreasing E-vortex and increasing A-vortex were observed in aging group without energy loss.
I read this paper with interest. Author showed visualization of intracardiac flow as a vortex formation by using vector flow mapping system and author also quantified vortex formation on basis of vortex morphological and vector analysis (i.e., circulation area and velocity). In addition, author calculated vortex kinetic analysis of intracardiac blood flow imaging. In this study, these analysis processes applied for younger and older patient groups for the purpose of investigation for the relationship among intracardiac vortex patterns, energetics, and aging heart. Author clearly showed the strong correlations between E-vortex and A-vortex from VEM analysis and previous parameters of diastolic function.
I read this paper with interest. This paper showed VFM might have a potential to visualize and quantify the intracardiac flow and dynamics. I would like to make a few comments.
Comments to Author
Major comments:
- In this study, age-related cardiac changes were evaluated by VFM analysis. Author should describe age distribution and average age in both younger and older age groups clearly. In addition, author should show the reason why author decided the age cut-off line in this grouping (younger <45 years, older > 45 years). Author should explain the study population in further detail. For example, average ages in both groups should be in Methods part and/or Table 1.
- Strong correlation between vortex flow analysis and other previous parameters (i.e., e’, a’). Author should describe the mechanism of these correlations clearly on discussion part. It could be help audiences understand the significance of these correlations.
- Author applied color Doppler flow mapping system for analysis of vortex flow in this study. On the other hand, there are different analysis methods for vortex flows (i.e., MRI, echocardiography with ultrasounds contrast agents). Author should add the information of superiority of CDFM on discussion part. These explanations could help the audience understand vector flow mapping system.
Minor comments:
- Introduction section should be more concise.
- Author should describe the correlations between the parameters from VFM and the other parameters for diastolic functions in both Results and Abstracts parts.
Please show this part in Results part:
E-vortex circulation correlated directly to e’ (p=0.004), 20 A-vortex circulation correlated directly to A and a’ (both p<0.0001), and S-vortex circulation corre-21 lated directly to s’ (p=0.028).
Author Response
Comments from Reviewer 4
Authors described the evaluation for the age-related cardiac changes, which were mainly diastolic dysfunction, by using vector flow mapping (VFM) analysis of intracardial flow.
Authors also calculated peak energy loss (EL) for comparison between young and aging hearts. Decreasing E-vortex and increasing A-vortex were observed in aging group without energy loss.
I read this paper with interest. Author showed visualization of intracardiac flow as a vortex formation by using vector flow mapping system and author also quantified vortex formation on basis of vortex morphological and vector analysis (i.e., circulation area and velocity). In addition, author calculated vortex kinetic analysis of intracardiac blood flow imaging. In this study, these analysis processes applied for younger and older patient groups for the purpose of investigation for the relationship among intracardiac vortex patterns, energetics, and aging heart. Author clearly showed the strong correlations between E-vortex and A-vortex from VEM analysis and previous parameters of diastolic function.
I read this paper with interest. This paper showed VFM might have a potential to visualize and quantify the intracardiac flow and dynamics. I would like to make a few comments.
Comments to Author
Major comments:
- In this study, age-related cardiac changes were evaluated by VFM analysis. Author should describe age distribution and average age in both younger and older age groups clearly. In addition, author should show the reason why author decided the age cut-off line in this grouping (younger <45 years, older > 45 years). Author should explain the study population in further detail. For example, average ages in both groups should be in Methods part and/or Table 1.
Authors’ response: Thank you for your comment. As per reviewer 1 and 2’s comments, we have now divided the cohort into 3 age groups by decades. We have also added the age and sex distributions in Table 1.
2. Strong correlation between vortex flow analysis and other previous parameters (i.e., e’, a’). Author should describe the mechanism of these correlations clearly on discussion part. It could be help audiences understand the significance of these correlations.
Authors’ response: Thank you for your comment. The mechanism for such correlation likely relates to stronger transmitral flow with stronger annular motion. We have added a statement in the relevant paragraph to highlight this (lines 189-192): “We also found significant correlations between vortex circulations and tissue doppler indices, which reflect that stronger mitral annular motions during LV filling and the ensuing stronger transmitral flow likely result in stronger vortical flow in the LV.”
3. Author applied color Doppler flow mapping system for analysis of vortex flow in this study. On the other hand, there are different analysis methods for vortex flows (i.e., MRI, echocardiography with ultrasounds contrast agents). Author should add the information of superiority of CDFM on discussion part. These explanations could help the audience understand vector flow mapping system.
Authors’ response: Thank you for your comment. We have added a paragraph in Discussion accordingly (lines 205-219): “Most previous studies of intracardiac flow have used other techniques, such as cardiac magnetic resonance and particle image velocimetry. Though extensively studied, these techniques each have significant limitations: cardiac magnetic resonance is expensive, requires significant time for processing which prohibits flow visualization at the time of imaging, and is limited in availability at many centres; meanwhile, particle image velocimetry, though allowing rapid flow visualization at the time of imaging, requires contrast agent injection, often turning an otherwise-non-invasive transthoracic echocardiography into an invasive procedure.[1] These may pose difficulties for researchers in recruiting research subjects, and more importantly, create major barriers for incorporating intracardiac flow imaging into clinical practice in the future. In contrast, VFM is inexpensive, completely non-invasive, and allows rapid flow visualization at the time of imaging. Future studies may therefore utilize VFM to investigate intracardiac flow in larger cohorts with wider age ranges and possibly follow-up studies, potentially better delineate the intracardiac flow alterations in aging and other conditions, as well as any prognostic significance of these alterations.”
Minor comments:
- Introduction section should be more concise.
Authors’ response: Thank you for your comment. We have shortened the introduction accordingly.
2. Author should describe the correlations between the parameters from VFM and the other parameters for diastolic functions in both Results and Abstracts parts.
Please show this part in Results part:
E-vortex circulation correlated directly to e’ (p=0.004), 20 A-vortex circulation correlated directly to A and a’ (both p<0.0001), and S-vortex circulation corre-21 lated directly to s’ (p=0.028).
Authors’ response: Thank you for your comment. We have added the results in lines 161-164: “E-vortex circulation correlated directly to e’ (rs=-0.297, p=0.003), A-vortex circulation correlated directly to A (rs=-0.606, p<0.0001) and a’ (rs=503, p<0.0001), and S-vortex circulation correlated directly to s’ (rs=-0.214, p=0.032).”
Round 2
Reviewer 1 Report
The authors have addressed all my comments, sometimes in an unexpected manner, by completely removing sections. The only remaining point is that I would have loved to see some reasoning around the issues in the final paper, at least concerning the limitations. Like for example a short statement/reference to why the method is considered to be less angle-dependent, or to why the anti-aliasing is considered to be so efficient, and maybe as pointed out in the first review, discuss the possible benefits of using high(er) frame rates.
Reviewer 4 Report
Manuscript has been improved. There was no comment for this version.
Author Response
Thank you very much for your positive comment. We also wish to thank you for providing your expert comments which substantially improved our manuscript.